# Augmentation of frontoparietal gamma-band phase coupling enhances human altruistic behavior

Jie Hu[1,2]☉*, Marius Moisa[2]☉, Christian C. Ruff[2,3,4]*

1 Shanghai Key Laboratory of Mental Health and Psychological Crisis Intervention, School of Psychology and Cognitive Science, East China Normal University, Shanghai, China, 2 Zurich Center for Neuroeconomics, Department of Economics, University of Zurich, Zurich, Switzerland, 3 Faculty of Medicine, University of Zurich, Zurich, Switzerland, 4 University Research Priority Program "Adaptive Brain Circuit Mechanisms in Development and Learning" (URPP AdaBD), University of Zurich, Zurich, Switzerland

☉ These authors contributed equally to this work.
* jhu@psy.ecnu.edu.cn (JH); christian.ruff@econ.uzh.ch (CCR)

## Abstract

Cooperation, productivity, and cohesion in human societies depend on altruism, the tendency to share resources with others even though this is costly. While altruism is a widely shared social norm, people vary strongly in their inclination to behave altruistically, in particular across situations with different types of inequality in resource distribution. What neurobiological factors underlie this variability? And can these be targeted by interventions to enhance altruistic behavior? Here, we build on electroencephalography (EEG) evidence that altruistic choices during disadvantageous inequality correlate with oscillatory gamma-band coherence between frontal regions (representing other's interest) and parietal regions (representing neural evidence accumulation). We apply a transcranial alternating current stimulation protocol designed to exogenously enhance this fronto-parietal coherence and find that this leads to increased altruism, specifically during disadvantageous inequality as hypothesized based on the EEG findings. Computational modeling reveals that this transcranial entrainment does not just add noise to the decision process but specifically increases the weight individuals assign to other-regarding concerns during choices. Our findings show that altruism can be enhanced by neurostimulation designed to enhance oscillatory synchronization between frontal and parietal areas. This establishes a neural basis for altruism and identifies a neural target for interventions aimed at improving prosocial behavior.

## Introduction

Altruism is the foundation for collaboration and solidarity in human society [1,2]. Lacking altruism is a hallmark of social deficits in psychiatric and neurological disorders

**Data availability statement:** All relevant data and analysis codes have been made publicly available on Open Science Framework (OSF, https://osf.io/g8m6p/). Data were analyzed using Rstudio (R Version 3.5.1) and Matlab 2023a (MathWorks).

**Funding:** This work received funding from the European Research Council (ERC) under the European Union's Horizon 2020 research and innovation programme (grant agreement No 725355, ERC consolidator grant BRAINCODES to C.C.R.), the University Research Priority Program 'Adaptive Brain Circuits in Development and Learning' (grant no. URPP AdaBD to C.C.R.) at the University of Zurich and the Swiss National Science Foundation (grant nos. 10006863 and 100019L-173248 to C.C.R.). This work has also received funding from the National Natural Science Foundation of China (grant nos. 32400881 and 32571237 to J.H.) and was supported by The Research Project of Shanghai Science and Technology Commission (20dz2260300 to J.H.) and The Fundamental Research Funds for the Central Universities. The funders had no role in study design, data collection and analysis, decision to publish, or preparation of the manuscript.

**Competing interests:** The authors have declared that no competing interests exist.

**Abbreviations:** ACC, anterior cingulate cortex; ADV, advantageous inequality condition; ANOVA, analysis of variance; BFs, Bayes factors; BIC, Bayesian Information Criterion; DIS, disadvantageous inequality condition; DLPFC, dorsolateral prefrontal cortex; EEG, electroencephalography; HD, high-definition; MNI, Montreal Neurological Institute; NIBS, noninvasive brain stimulation; PFC, prefrontal cortex; rIPL, inferior parietal lobe; SNS-Lab, Laboratory for Social and Neural Systems; SSM, sequential sampling models; tACS, transcranial alternating current stimulation; tDCS, transcranial direct current stimulation; TMS, transcranial magnetic stimulation; TPJ, temporoparietal junction; VMPFC, ventromedial prefrontal cortex; ZNE, Zurich Center for Neuroeconomics.

(e.g., psychopathy, autism, and alexithymia) [3,4] and lies at the heart of many collective social problems (e.g., high crime rate and poor crisis management) [5,6]. The neural basis of altruism is therefore intensely studied in neuroscience, with the aim to identify neural mechanisms that may be targeted by interventions to enhance altruistic behavior.

This research has identified various brain regions in frontal and parietal cortex that may play critical roles in social decision-making [7–10]. Studies using fMRI have shown that activity in areas such as frontal cortex (e.g., ventral and dorsal prefrontal cortex (PFC), and dorsal anterior cingulate cortex) and parietal cortex (e.g., temporoparietal junction (TPJ) and inferior parietal cortex) is engaged in neural processing of other-interest versus self-interest, context-dependent social signals (e.g., equality), and in processes that arbitrate between different social motives (e.g., altruistic versus selfish choice) [8,9,11,12]. Importantly, the evidence suggests that distinct regions of the prefrontal cortex may contribute differently to self- versus other-oriented behavior: While the ventromedial prefrontal cortex (VMPFC) has been consistently implicated in representing the subjective value of prosocial options and integrating other-regarding preferences [13,14], the dorsolateral prefrontal cortex (DLPFC) has been more strongly associated with implementing control processes that may either promote self-interest or support norm compliance and fairness depending on contextual demands [15,16]. As a third region, the anterior cingulate cortex (ACC) has been shown to monitor conflicts between self- and other-regarding motives, and to track the cost of prosocial acts [17]. Moreover, when people take altruistic choices, neural responses in medial prefrontal cortex and TPJ are stronger than during selfish choices [9,12]. While fMRI studies allow for such region-specific insights, electroencephalography (EEG) provides more limited spatial resolution. Nevertheless, prior EEG findings from our lab suggest that central prefrontal and parietal activity is particularly involved in weighing other-related interests, making it unlikely that these effects primarily reflect DLPFC function [18]. Together, these findings highlight that altruistic decision-making cannot be attributed to a single frontal locus but instead relies on partially dissociable and interacting prefrontal processes, which form functional networks with parietal regions [9,11,13].

Adding to the fMRI evidence, but extending it in terms of causality, brain stimulation research has provided substantial evidence that exogenously enhancing or disrupting neural activity in specific regions can directly modulate altruistic behavior. For instance, noninvasive stimulation studies using transcranial direct current stimulation (tDCS) and transcranial magnetic stimulation (TMS) have suggested distinct yet complementary roles of prefrontal and parietal cortices in altruistic behavior. On the one hand, enhancing DLPFC activity - which is involved in resolution of conflict between self-related interest and other-regarding interest - has been shown to suppress selfish motives and facilitate honesty [19]. Dampening DLPFC activity, by contrast, was shown to promote selfish motives and reduce altruistic choices [11]. On the other hand, modulating activity in TPJ and inferior parietal lobe (rIPL)—regions implicated in perspective-taking and other-interest representation—can alter empathic concern and altruistic giving, either promoting or impeding altruistic choices

[11,20,21]. Together, these findings provide convergent causal evidence that specific cortical regions (i.e., prefrontal and parietal regions) play distinct and complementary roles in guiding altruistic preferences, bridging correlational neuroimaging results with mechanistic insights into the neural bases of human altruism.

Beyond the local activity of specific brain regions, however, recent studies have highlighted the role of large-scale inter-regional connectivity in social decision-making [22–24]. For example, stronger connectivity between frontal (e.g., medial/lateral prefrontal cortex) and parietal cortex (e.g., inferior parietal cortex or TPJ) was shown to correlate with greater pro-social preferences [11,13]. One potential mechanistic explanation for the role of such fronto-parietal interactions in social decision-making is based on studies of nonsocial value-based decision-making, which suggest that choice evidence derived from value computation and comparison in medial prefrontal cortex is communicated to parietal cortex where it is transformed into response signals [25–27]. For example, greater phase-coupling between frontal and parietal regions—correlated to evidence accumulation—is associated with better performance in making precise value-based decisions [28,29]. In line with these suggestions, in a recent EEG study, we revealed that during altruistic choices, neural activity in a medial frontal region is involved in processing of other' interest, while a parietal region is involved in mapping value evidence to motor responses. Importantly, the strength of phase-coupling between these two regions (i.e., fronto-parietal synchronization) was positively associated with individuals' altruistic preferences, whereas no such relationship was found for the local strength of oscillations in these brain regions (see S1 Fig for supplementary analyses of our previous EEG data). This suggests that tighter functional communication of these two areas may be associated with a larger impact of other-regarding concerns on choice, by stronger integration of other-regarding concern processed in medial frontal region into the final decision process encoded in the parietal region [18].

Since interregional phase-coupling of neural oscillations is one way by which activity in one neural population can impact on activity in other connected neural populations [30,31], the previous findings thus implicate that the phase-coupling within the fronto-parietal circuitry may support the inter-regional neural communication underlying both nonsocial and social decision-making [18,28,32]. Nevertheless, the previous neuroimaging studies (fMRI and EEG) could only provide correlative evidence for this relationship between fronto-parietal interaction and social behaviors [11,13,33,34]; very few studies have established the causal necessity of inter-regional connectivity for social decision-making [35]. It is therefore still largely unknown whether fronto-parietal functional interactions are causally involved in modulating individuals' social preferences and behaviors, or whether this is only an epiphenomenon of the decision process.

Beyond correlative evidence for the general functional importance of inter-regional connectivity, recent advances in noninvasive brain stimulation (NIBS) protocols have provided causal tests of the "communication through coherence" hypothesis [36]. For instance, paired stimulation approaches using TMS stimulators have been shown to selectively manipulate inter-areal coherence between premotor and motor cortices, thereby altering inter-regional communication [37]. These studies highlight the feasibility of experimentally modulating neural coherence to probe its causal role in cognition and behavior. Consistent with the general rationale underlying this line of work, we here employ high-definition tACS to target fronto-parietal synchronization, to directly examine whether exogenous modulation of frequency-specific long-range coherence can shape altruistic decision-making.

One factor that complicates the study of the neural basis of altruism is that altruistic decisions can be driven by distinct motives in different contexts [18,38–40]. For instance, any inequality in resources between two individuals can substantially affect wealth distribution choices: when people possess more than others (advantageous inequality, ADV), stronger "guilt" emotion may drive them to share more; whereas when people possess less than others (disadvantageous inequality, DIS), "envy" emotion may drive them to share less [12,23,41–44]. Recent brain stimulation studies have suggested distinct causal roles of frontal and parietal regions in modulating such context-dependent altruistic preferences. While enhancing activity of parietal regions (e.g., the TPJ) was suggested to promote altruistic behaviors specifically in advantageous inequality contexts [20,21,45], manipulating oscillation of frontal regions (e.g., lateral prefrontal cortex) was found to enhance individuals' preferences for unequal choices specifically in disadvantageous inequality contexts [46]. However,

these studies only linked activity of a single region to altruistic preferences, thereby not addressing how large-scale interregional communications (i.e., within the frontoparietal network) can causally affect altruism in different contexts. Our previous study suggested that while in both these contexts, similar parietal evidence accumulation signals could be observed in the EEG signal, the strength of frontoparietal phase-coupling was correlated with altruistic preferences mainly during disadvantageous inequality [18]. However, it is unclear whether this frontoparietal oscillatory coherence evident in the EEG signal is linked to altruism in a causal sense—either by implementing a general tendency or a specific motive underlying altruism in a specific context.

In the current study, we address these issues by combining a modified dictator game with a high-definition (HD) transcranial alternating current stimulation (tACS) approach designed to exogenously modulate phase coupling of distant cortical brain regions [47,48]. This approach allows us to clarify three fundamental questions regarding the role of interregional phase-coupling in altruistic decision-making. First: is it in principle possible to enhance altruism by external stimulation designed to affect neural coherence? Second: is it specifically gamma-band frontoparietal synchronization that is causally relevant for altruistic choices? Third: is gamma-band frontoparietal synchronization relevant for altruistic motivation in general, or just for specific motives present during disadvantageous (versus advantageous) inequality?

Based on prior evidence linking gamma-band oscillations to value integration and evidence accumulation, we expected that enhancing gamma-band synchronization between frontal and parietal regions would promote altruism, whereas stimulation in other frequency bands (i.e., alpha band) would not generate such effects. Moreover, we anticipated that this facilitation would be most pronounced in contexts of disadvantageous inequality, since the strength of frontoparietal phase-coupling was correlated with altruistic preferences mainly during disadvantageous inequality.

## Results

Participants played as proposers in a modified Dictator Game and had to choose between two possible monetary distributions between themselves and anonymous partners (Fig 1A). As in our previous study, we varied the sizes of payoffs in a way that created two different inequality contexts—disadvantageous (DIS) in which participants got less than their partners for both distribution options and advantageous (ADV) in which participants got more than their partners for both distribution options (for detailed trial set distribution see S2 Fig). These two types of trials were only defined by the size of the payoffs presented on the screen (i.e., there were no other visual markers) and were randomly intermixed in a manner that led to strongly varying inequality and context variation from trial to trial, while still ensuring an overall balanced presentation across the whole experiment (see S2 Fig).

During the task, participants received tACS by means of a high-definition tACS setup designed to noninvasively synchronize cortical rhythms at a specific frequency between the frontal and parietal regions underlying the electrodes [47,48]. To this end, we placed two arrays of small electrodes (3 × 1 HD electrode montage) over the specific positions where we had previously observed the coherence pattern [18] and provided focal electrical current stimulation to entrain the coherence between the corresponding frontal and parietal areas (Fig 1B). Participants received three different types of entrainment: gamma, alpha, and sham stimulation (Fig 1C and 1D). Importantly, the tACS oscillations were temporally aligned between the two arrays of electrodes (i.e., in-phase stimulation, phase difference = 0). We only included synchronous stimulation (in-phase entrainment) and not asynchronous stimulation (anti-phase entrainment) over the two regions, because the simulated electric fields induced by synchronous stimulation over the two regions are clearly separate from each other (Fig 1B), whereas the simulated electric fields induced by asynchronous stimulation over the two regions may overlap with each other (S3 Fig). Therefore, anti-phase entrainment tACS would have been ambiguous with respect to the origin of any effect, which may reflect either a local oscillation change in the region between the two targeted sites or inter-regional synchronization between the targeted frontal and parietal regions.

To better understand which cortical regions were targeted by our tACS montage, we simulated electric field distributions and mapped them onto MNI coordinates corresponding to regions previously implicated (by fMRI studies) in altruistic

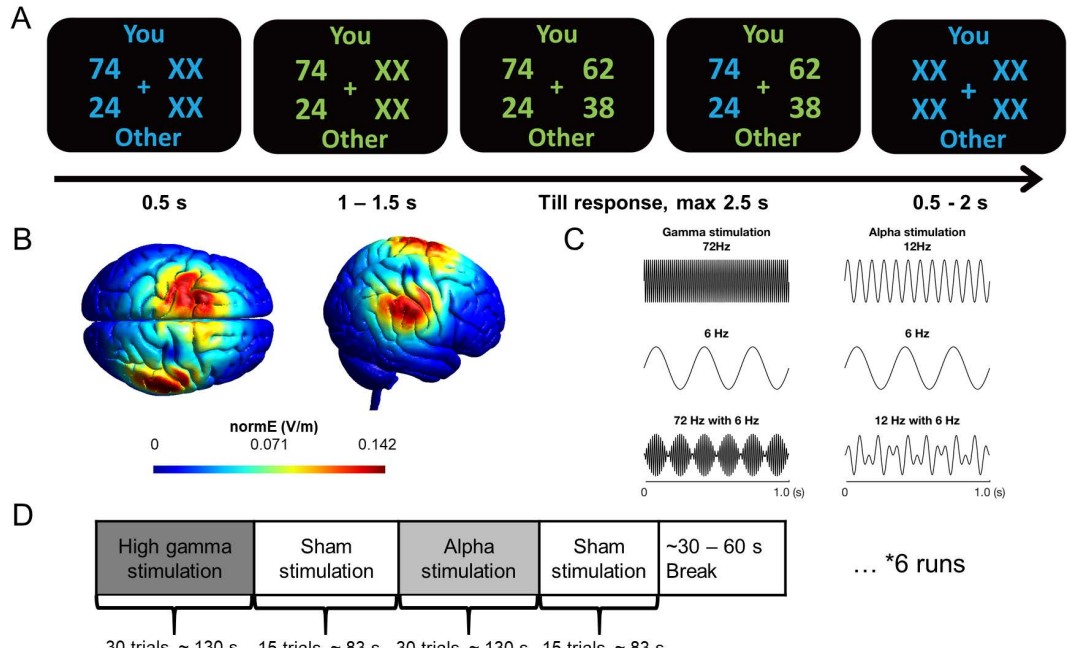

**Fig 1. Experimental design and protocol. (A)** Example of a single trial. Participants acted as proposers to distribute a certain amount of monetary tokens between themselves and anonymous partners. In the beginning of each trial, participants were presented with one reference option in blue for 0.5 s. The trial starts when the font color changes from blue to green. After the presentation of the 2nd option, participants needed to choose between the two options within 2.5 s. The selected option was highlighted in blue before the inter-trial interval. Font color assignment to phases (i.e., blue and green to response) was counterbalanced across participants. **(B)** tACS electric field density simulation. Two sets of small electrodes (3 × 1 HD electrode montage) were mounted over frontal and parietal regions which were located based on our previous EEG study [18]. The normalized simulated electric field distribution showed that the targeted parietal and frontal areas under the two sets of tACS electrodes are affected by the stimulation with a relatively good spatial focality. **(C)** Schematic of the alternating currents delivered to participants. Participants received alternating currents at 72 Hz with an envelope of 6 Hz (gamma-band stimulation, left bottom panel), alternating currents at 12 Hz with an envelope of 6 Hz (alpha-band stimulation, right bottom panel) as active control stimulation, and sham stimulation. **(D)** Participants received these three types of stimulation during the task. Each run lasted ~7 min and the order of the stimulation blocks was counterbalanced across different runs.

and other-regarding decision-making [9,11,12,49,50], or found to be involved in "social interaction" based on Neurosynth meta-analysis (Table 1).

Simulations were performed using SimNIBS with 3.2 mA peak-to-peak stimulation intensity, see Methods section for more details [51,52]. The results indicated that the electric field strength beneath the two main electrodes reached ~ 0.13–0.14 V/m, which was substantially higher than in most other cortical regions examined. Notably, the central parietal electrode was positioned directly over the right IPL (57, −37, 52; bold in Table 1), a region previously implicated in other-interest processing [11], where the mean field strength reached 0.113 V/m (max = 0.145 V/m). Two nearby parietal sites corresponding to the right TPJ (63, −42, 21; 51, −51, 27; bold in Table 1) also exhibited moderate stimulation levels (mean = 0.075 and 0.072 V/m, respectively), consistent with prior findings that link TPJ activity to other-regarding preferences [9,12].

By contrast, most frontal regions received less than half of the field strength observed beneath the central electrodes. Only the left superior frontal gyrus (SFG; −22, 19, 60; bold in Table 1) exhibited a moderate electric field (mean = 0.077 V/m), aligning with prior reports of its role in other-interest processing [9].

Overall, these additional analyses demonstrate that the stimulated regions overlapped with key nodes previously implicated by fMRI studies in other-regarding and altruistic decision-making networks, thereby reinforcing the functional

**Table 1. Simulated electric field intensities at regions of interest derived from prior studies on other-regarding preferences and altruistic decision-making.**

| Regions | Peak coordinates | | | Contrasts/key term | Mean intensity in a sphere of 8 mm radius (V/m) | References |
|---|---|---|---|---|---|---|
| | x | y | z | | | |
| **Prefrontal cortices** | | | | | | |
| R ACC | 9 | 36 | 3 | Other-interest | 0.0162 | [9] |
| R MCC | 1 | 9 | 30 | Other-interest ∩ Self-interest | 0.0410 | |
| **L SFG** | **−22** | **19** | **60** | | **0.0770** | |
| R VMPFC | 6 | 38 | 9 | | 0.0185 | |
| R DLPFC | 51 | 11 | 22 | | 0.0309 | [11] |
| L DLPFC | −40 | 38 | 23 | Unfair > Fair | 0.0230 | [49,50] |
| R DLPFC | 40 | 37 | 22 | | 0.0208 | |
| R ACC | 4 | 19 | 39 | | 0.0426 | [50] |
| L ACC | −9 | 26 | 28 | | 0.0320 | |
| R MFG | 34 | 23 | 35 | | 0.0370 | |
| L MFG | −39 | 21 | 41 | | 0.0538 | |
| L VMPFC | −4 | 44 | −8 | Social interaction | 0.0104 | Meta-analyses in Neurosynth |
| R DMPFC | 2 | 56 | 16 | | 0.0137 | |
| R MCC | 2 | 26 | 42 | | 0.0427 | |
| L DLPFC | −44 | 18 | 24 | | 0.0256 | |
| R DLPFC | 50 | 10 | 22 | | 0.0330 | |
| L IFG | −48 | 30 | −8 | | 0.0088 | |
| R IFG | 52 | 30 | 4 | | 0.0147 | |
| **Parietal cortices** | | | | | | |
| **R TPJ** | **63** | **−42** | **21** | Altruistic preference/ decision | **0.0747** | [12] |
| | 60 | −44 | 18 | | 0.0598 | |
| | **51** | **−51** | **27** | Other-interest > Self-interest | **0.0715** | [9] |
| | 39 | −63 | 21 | Other-interest | 0.0440 | |
| **R IPL** | **57** | **−37** | **52** | | **0.1132** | [11] |
| R TPJ | 58 | −42 | 12 | Social interaction | 0.0459 | Meta-analyses in Neurosynth |
| L TPJ | −50 | −60 | 24 | | 0.0182 | |

Abbreviations: ACC, Anterior cingulate cortex; MCC, Middle cingulate cortex; SFG, Superior frontal gyrus; VMPFC, Ventromedial prefrontal cortex; DLPFC, Dorsolateral prefrontal cortex; DMPFC, Dorsomedial prefrontal cortex; IFG, Inferior frontal gyrus; MFG, Middle frontal gyrus; TPJ, Temporoparietal junction; IPL, Inferior parietal lobe; R, Right; L, Left.

relevance of our tACS montage. The present findings extend beyond our previous EEG-based results, by highlighting the anatomical and functional characteristics of the targeted brain regions.

We included alpha entrainment as an active control condition for potentially nonspecific effects of the stimulation [53]. Since our previous study had found that greater altruistic preferences correlate with stronger frontoparietal synchronization in the gamma-band frequency [18], we expected that only gamma, but not alpha entrainment, would enhance altruistic preferences. For the sham stimulation, we provided either gamma (for the sham block following the gamma entrainment block) or alpha entrainment (for the sham block following the alpha entrainment block) for only 2 s every 27 s during the two sham stimulation blocks within each experimental run. The order of the entrainment type (i.e., gamma or alpha) for the two sham stimulation blocks was counterbalanced across different experimental runs.

To examine how individuals' altruistic behavior was influenced by the different types of stimulation, we performed linear mixed-effects regression of altruistic choice (more precisely, the probability of choosing the more altruistic option on each trial) on inequality context (ADV versus DIS), stimulation type (gamma versus alpha, gamma versus sham, and alpha versus sham), and their interactions. In line with our hypothesis, we found that gamma entrainment indeed led to an increase in the probability of choosing the altruistic option (0.16 ± 0.02, Mean ± SE), compared to both alpha entrainment (0.14 ± 0.02, main effect of gamma entrainment in Table 2, Model 1: $\beta$ = 0.18 ± 0.10, Estimate ± SE, $p$ = 0.032) and sham stimulation (0.15 ± 0.02, main effect of gamma entrainment in Table 2, Model 2: $\beta$ = 0.19 ± 0.10, $p$ = 0.028). Altruistic choice probability did not differ between alpha entrainment and sham stimulation [main effects of sham (alpha) stimulation in Table 2, Model 1 (Model 2): $\beta$ = −0.01 ± 0.10 ($\beta$ = 0.01 ± 0.10), $p$ = 0.942]. This shows that fronto-parietal gamma-band coherence is indeed causally relevant for altruistic behavior, and that it is possible to enhance altruism by strengthening this coherence.

As in our previous EEG study, the probability of altruistic choice was higher in the ADV (0.19 ± 0.03) than in the DIS context (0.11 ± 0.02, main effect of inequality context in Table 2, Model 1: $\beta$ = 0.82 ± 0.09, $p$ < 0.001; in Table 2, Model 2: $\beta$ = 0.89 ± 0.09, $p$ < 0.001), consistent with previous suggestions that individuals consider others' interest more in their choices when they possess more compared to when they possess less than others [18]. As we stimulated each participant with an electric current intensity in line with his/her tolerance level for the stimulation currents tested before the experiment runs, the peak-to-peak current intensity for different participants varied between 2.4 and 4 mA. Therefore, in the regression analyses, we also included individual electric current intensity for each participant and individual discomfort ratings for each stimulation run as control variables. The effects reported above were not influenced by potential differences in

**Table 2. Logistic mixed-effects model results of choice data.**

| Fixed effects | Model 1 | | Model 2 | |
|---|---|---|---|---|
| | β (95% CI) | *p*-value | β (95% CI) | *p*-value |
| Intercept | −2.89*** (−3.36 to −2.42) | <0.001 | −2.90*** (−3.36 to −2.43) | <0.001 |
| Inequality context (C) | 0.82*** (0.63–1.00) | <0.001 | 0.89*** (0.71–1.08) | <0.001 |
| Gamma (G) | 0.18* (−0.01 to 0.38) | 0.032 | 0.19* (−0.004 to 0.39) | 0.028 |
| Sham (S) | −0.01 (−0.21 to 0.19) | 0.942 | − | − |
| Alpha (A) | − | − | 0.01 (−0.19 to 0.21) | 0.942 |
| G*C | −0.07 (−0.32 to 0.19) | 0.604 | −0.14 (−0.40 to 0.11) | 0.272 |
| S*C | 0.08 (−0.18 to 0.34) | 0.570 | − | − |
| A*C | − | − | −0.08 (−0.34 to 0.18) | 0.570 |
| Conditional R² | 0.42 | | 0.42 | |
| LL | −4,655 | | −4,655 | |
| BIC | 9,377 | | 9,377 | |

One-tailed tests for the effects of "Inequality context" (C) and "Gamma" entrainment (G), but two-tailed statistics for the other effects with undirected hypotheses. Gamma (G): gamma entrainment; Sham (S): sham stimulation; Alpha (A): alpha entrainment; LL: log-likelihood; BIC: Bayesian Information Criterion. *** $p$ < 0.001; ** $p$ < 0.01; * $p$ < 0.05.

discomfort experienced by the participants for different types of entrainments, or by the individual electric current intensity the participants received (Table 3, Models 1 and 2, see Methods).

Although the interaction between stimulation type (gamma versus alpha or gamma versus sham) and inequality context was not statistically significant, inspection of the data in the two inequality contexts showed that the main effect of gamma entrainment appeared to mainly reflect the increasing effect of gamma entrainment on altruistic choice in the DIS context (probability of altruistic choice in gamma: 0.12 ± 0.02; versus in sham: 0.10 ± 0.02, $t$(43) = 2.07, $p$ = 0.044, *Cohen's d* = 0.31; versus in alpha: 0.10 ± 0.02, $t$(43) = 2.27, $p$ = 0.028, *Cohen's d* = 0.34). In the ADV context, no such effects were evident (probability of altruistic choice in gamma: 0.20 ± 0.03, in alpha: 0.19 ± 0.03, in sham: 0.20 ± 0.03, all $p$s > 0.09) (Fig 2A and 2B; S1 and S2 Tables). This suggests that fronto-parietal gamma-band coherence may not be equally important for all motives underlying altruistic choice, but may particularly affect those motives guiding choices during disadvantageous inequality.

How exactly does gamma entrainment promote altruistic choice? In principle, altruistic choices could increase because the stimulation enhances the influence of other-regarding concerns on behavior, in line with the notion that the stimulation increases how behavior is guided by altruistic motives. Alternatively, the stimulation could impact on more general aspects of the choice process that nevertheless can alter the apparent relative impacts of selfish and other-regarding concerns

**Table 3. Logistic mixed-effects model results of choice data.**

| | Model 1 | | Model 2 | |
|---|---|---|---|---|
| **Fixed effects** | **β (95% CI)** | **p-value** | **β (95% CI)** | **p-value** |
| Intercept | −2.89*** (−3.36 to −2.42) | <0.001 | −2.89*** (−3.36 to −2.42) | <0.001 |
| Inequality context (C) | 0.82*** (0.63–1.00) | <0.001 | 0.89*** (0.71–1.08) | <0.001 |
| Gamma (G) | 0.19* (−0.02 to 0.39) | 0.036 | 0.19* (−0.009 to 0.39) | 0.030 |
| Sham (S) | −0.002 (−0.22 to 0.22) | 0.987 | – | – |
| Alpha (A) | – | – | 0.002 (−0.22 to 0.22) | 0.987 |
| G*C | −0.07 (−0.32 to 0.19) | 0.603 | −0.14 (−0.40 to 0.11) | 0.273 |
| S*C | 0.08 (−0.18 to 0.34) | 0.570 | – | – |
| A*C | – | – | −0.08 (−0.34 to 0.18) | 0.570 |
| Discomfort rating | 0.01 (−0.16 to 0.18) | 0.906 | 0.01 (−0.16 to 0.18) | 0.906 |
| Intensity | 0.12 (−0.33 to 0.57) | 0.598 | 0.12 (−0.33 to 0.57) | 0.598 |
| Conditional R² | 0.42 | | 0.42 | |
| LL | −4,655 | | −4,655 | |
| BIC | 9,396 | | 9,396 | |

One-tailed tests for the effects of "Inequality context" (C) and "Gamma" entrainment (G), but two-tailed statistics for the other effects with undirected hypotheses. Discomfort rating, participants rated their discomfort due to the stimulation after each entrainment/stimulation block. Intensity, each participant was stimulated with an electric current intensity on his/her tolerance level for the stimulation currents tested before the experiment runs. Gamma (G): gamma entrainment; Sham (S): sham stimulation; Alpha (A): alpha entrainment; LL: log-likelihood; BIC: Bayesian Information Criterion. *** $p$ < 0.001; ** $p$ < 0.01; * $p$ < 0.05.

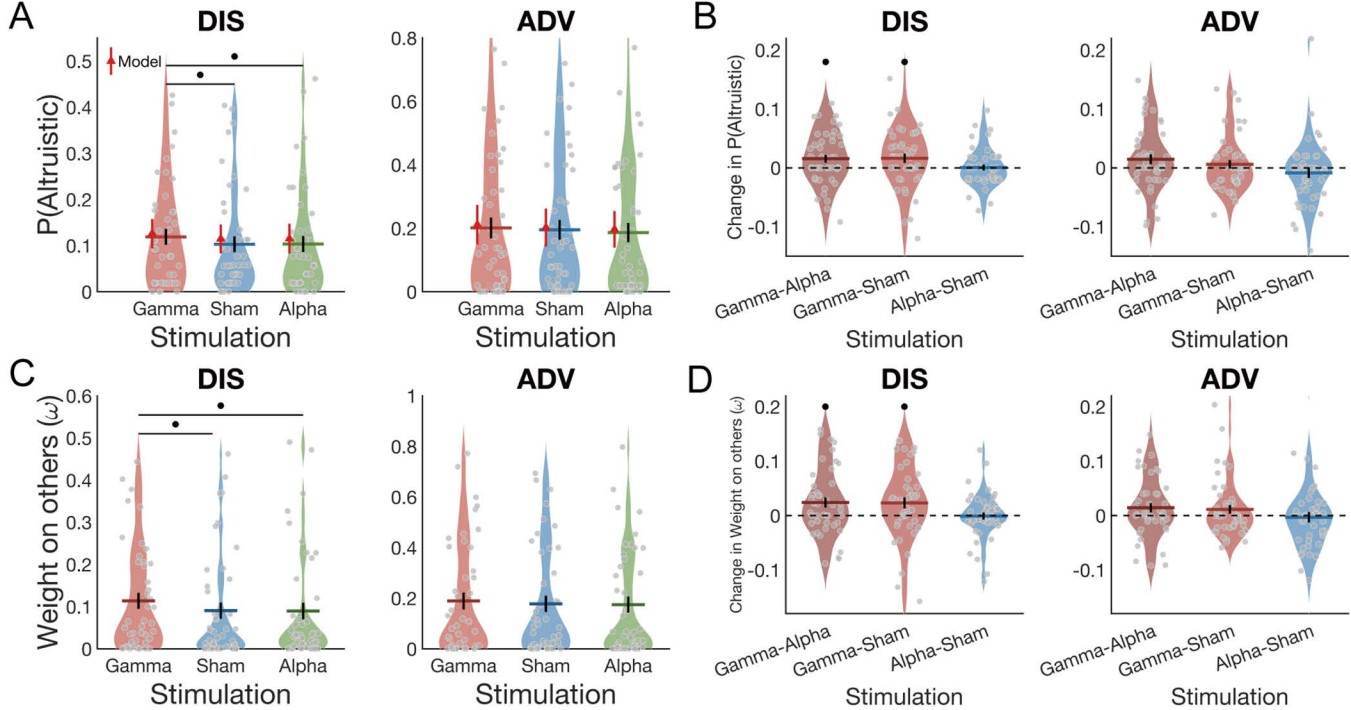

**Fig 2. Behavioral and modeling results. (A)** The probability of altruistic choice in DIS context was higher during gamma entrainment relative to both sham and alpha entrainment. The winning model recovers the effects of stimulation. Colored bars indicate the means and black error bars display the standard errors of the mean (SE). The red triangle dots and error bars represent the model simulation mean and 95% confidence intervals (CIs). **(B)** The difference in the probability of altruistic choice in the DIS context was significantly higher than 0 for gamma entrainment compared to both alpha entrainment and sham stimulation. **(C)** The winning model shows that gamma entrainment, relative to both sham and alpha entrainment, increased the weight on others in the DIS context. **(D)** The difference in the parameter of weight on others in the DIS context was significantly higher than 0 for gamma entrainment compared to both alpha entrainment and sham stimulation. Colored bars indicate the means for each stimulation type, and black error bars display the standard errors of the mean (SE). Each gray dot represents one participant. •, $p < 0.05$. The data underlying this figure can be found in https://osf.io/g8m6p/.

[20,54]. For instance, the stimulation could change the consistency of choices with (altruistic) decision values [54], or it could alter the decision weights people place on the efficiency of the monetary distributions (i.e., on the overall benefit for both parties, irrespective of how this is distributed) [20].

To test which of these possible effects was most evident in our data, we implemented computational modeling analyses. These were examined with comparison of three alternative models how our tACS entrainment affected different latent model variables corresponding to these different aspects of the choice process. In all the alternative models, we captured how each individual weighed others' interest versus their own interest to construct an integrated decision utility. To also assess consistency with the computed integrated decision values, we included either a single inverse temperature parameter in model M1 or condition-specific inverse temperature parameters in model M2. In a third model (M3), we additionally tested the possibility that individuals' decisions may depend not only on their own and other's interest but also on the overall benefit for both parties (i.e., efficiency).

Specifically, in M1, the subjective value difference between the more generous option and the more selfish option (VD) was constructed based on the Charness-Rabin utility model [41] in which people weigh between outcomes for themselves and for others (for detailed model specification and construction, see Methods section). We calculated the trial-by-trial value difference (VD) between the two options and employed a softmax function with a single inverse temperature

 

parameter to transform these value differences into probabilities of choosing the more altruistic option in different conditions. In M2, we included six inverse temperature parameters for the six different conditions. In M3, we included a set of parameters to measure to what extent individuals' decisions were also based on efficiency considerations.

Model comparison analysis showed that the first model (M1) outperformed the other models (S3 Table). M1 had the lowest BIC value among the three models and the Bayes factors (BFs) relative to the alternative models were over 100, indicating "very strong" evidence that M1 is superior to the other models. Posterior predictive checks showed that the altruistic choice probabilities predicted by the fitted winning model indeed captured the observed choice probabilities well (S4 Fig); moreover, the same logistic mixed-effects regression of these simulated choices revealed similar effects as participants' real choices. Specifically, the probability of altruistic choice was increased by gamma entrainment, both relative to alpha entrainment (main effect of gamma entrainment in Table 4, Model 1: $\beta = 0.23 \pm 0.10$, $p = 0.008$) and relative to sham stimulation (main effect of gamma entrainment in Table 4, Model 2: $\beta = 0.24 \pm 0.10$, $p = 0.007$). We also performed parameter recovery analyses to validate the winning model. The results showed that we could reliably recover the parameters in M1 (S5 Fig), confirming that the winning model captures the specific motives and choice processes underlying participants' altruistic behavior.

To confirm that gamma-band frontoparietal phase-coupling indeed increased individuals' altruistic preferences (i.e., their decision weight over others' interest, $\omega$), we analyzed how the tACS changed the winning model's parameters. We did so with analysis of variance (ANOVA) in the M1 altruistic preference parameter $\omega$, using tACS entrainment and inequality context as the independent variables. This confirmed that the stimulation changed altruistic preference, with a significant main effect of entrainment ($F(2, 86) = 4.56$, $p = 0.013$, $\eta^2_{partial} = 0.10$) and a significant main effect of inequality context ($F(1, 43) = 10.31$, $p = 0.003$, $\eta^2_{partial} = 0.19$), but an insignificant interaction effect between entrainment and

**Table 4. Logistic mixed-effects model results of simulated choice data.**

| | Model 1 | | Model 2 | |
|---|---|---|---|---|
| **Fixed effects** | **β(95% CI)** | **p-value** | **β (95% CI)** | **p-value** |
| Intercept | −2.71*** (−3.13 to −2.29) | <0.001 | −2.71*** (−3.13 to −2.29) | <0.001 |
| Inequality context (C) | 0.84*** (0.66–1.02) | <0.001 | 0.94*** (0.76–1.11) | <0.001 |
| Gamma (G) | 0.23** (0.04–0.42) | 0.008 | 0.24** (0.05–0.43) | 0.007 |
| Sham (S) | −0.008 (−0.20 to 0.19) | 0.937 | – | – |
| Alpha (A) | – | – | 0.01 (−0.19 to 0.20) | 0.937 |
| G*C | −0.12 (−0.37 to 0.12) | 0.324 | −0.22 (−0.47 to 0.03) | 0.081 |
| S*C | 0.10 (−0.16 to 0.35) | 0.459 | – | – |
| A*C | – | – | −0.10 (−0.35 to 0.16) | 0.459 |
| Conditional R² | 0.37 | | 0.37 | |
| LL | −4,961 | | −4,961 | |
| BIC | 9,988 | | 9,988 | |

One-tailed tests for the effects of "Inequality context" (C) and "Gamma" entrainment (G), but two-tailed statistics for the other effects with undirected hypotheses. Gamma (G): gamma entrainment; Sham (S): sham stimulation; Alpha (A): alpha entrainment; LL: log-likelihood; BIC: Bayesian Information Criterion. *** $p < 0.001$; ** $p < 0.01$; * $p < 0.05$.

inequality context ($F$(2, 86) = 0.72, $p$ = 0.490, $\eta^2_{partial}$ = 0.02). Post-hoc analyses confirmed that individuals' altruistic preferences were increased by gamma entrainment (0.15 ± 0.02) relative to both alpha entrainment (0.13 ± 0.02, $p$ = 0.005, *Cohen's d* = 0.41) and sham stimulation (0.13 ± 0.02, $p$ = 0.013, *Cohen's d* = 0.37). Again, this effect was specific to gamma stimulation, as there were no significant differences in altruistic preferences between alpha entrainment (0.13 ± 0.02) and sham stimulation (0.13 ± 0.02, $p$ = 0.386, *Cohen's d* = 0.04). Finally, and again consistent with the model-free analyses, individuals' altruistic preferences were stronger during advantageous (ADV, 0.18 ± 0.03) than disadvantageous inequality (DIS, 0.10 ± 0.02, $p$ = 0.002, *Cohen's d* = 0.48), and only in the DIS context were the individuals' weights on other-interest significantly increased by gamma entrainment (0.11 ± 0.02; both relative to sham (0.09 ± 0.02), $t$(43) = 2.28, $p$ = 0.042, *Cohen's d* = 0.34 and to alpha (0.09 ± 0.02), $t$(43) = 2.59, $p$ = 0.020, *Cohen's d* = 0.39). The corresponding effects in the ADV context were small and all failed to reach significance (gamma entrainment: 0.19 ± 0.03, alpha entrainment: 0.17 ± 0.03, sham simulation: 0.18 ± 0.03, $p$s > 0.1) (Fig 2C and 2D). The modeling analyses therefore confirmed that gamma entrainment increased altruistic concerns, in a manner that was more marked during situations with disadvantageous inequality where participants had less than the other participant to begin with.

## Discussion

It is widely acknowledged that a lack of altruism leads to greater problems for individuals' interpersonal interactions and for human society in general [3–6]. Therefore, clarifying the neural basis of altruism and developing efficient interventions to enhance altruism has become an important issue. Although previous studies have explored putative neural mechanisms by which our brains may take altruistic decisions (e.g., specific patterns of brain activity and/or connectivity correlating with valuation and evidence accumulation) [9,13,18], there is still little evidence that neural mechanisms supporting functional interactions between brain areas are in fact causally relevant for altruistic behavior [35,55]. To fill in this gap, our study provides evidence that altruism can be substantially enhanced by exogenously increasing gamma-band frontoparietal oscillatory synchronization.

Our results show that enhancing frontoparietal coherence in the gamma-band leads to more altruistic choices by strengthening altruistic preferences, suggesting that more efficient information sharing and integration of other-interest between the frontal and parietal cortex may promote more prosocial behavior. This interpretation is consistent with findings that frontal EEG responses to other-interest are correlated with individuals' altruistic preferences [18], and with the well-documented role of parietal regions in evidence accumulation processes underlying nonsocial [32,56] and social decision-making [18,27,28,57–59]. Our causal results integrate these separate sets of observations and suggest that frontal regions implement signals that evaluate others' interests, which are communicated to parietal regions where evidence is integrated to guide the appropriate decisions. Tighter integration of these two regions during stimulation may therefore lead to a larger impact of other-regarding concerns on choice, by stronger integration of other-regarding concerns into the final decision. Our findings thus suggest a specific brain network mechanism by which other-interest is incorporated into the choice process, the expression of which may plausibly underlie the strength of an individuals' altruistic preferences.

Importantly, such a mechanistic process is consistent with empirical patterns of both functional [25,27] and anatomical connectivity [32] between frontal and parietal regions. Note that a very similar logic has been used to interpret the functional role of increases in functional connectivity between VMPFC and TPJ during directed attention to other's payoffs [8,13]; our findings are thus consistent with general ideas about the role of functional communication between brain regions for altruism. However, our results provide causal evidence that it is specifically frontoparietal gamma synchronization that is causally required for altruistic choice, consistent with proposals that this specific neurophysiological mechanism may provide a neural pathway for information sharing and integration underlying diverse types of decision-making [26–29].

Consistent with these proposals, our previous EEG study had suggested that it is the inter-regional synchronization between frontal and parietal regions, rather than local activity/oscillations in frontal or parietal regions, that correlates with individuals' altruistic preferences [18]. To confirm this even more clearly, we re-analyzed our previous EEG data to

examine whether the local oscillations of frontal and parietal regions are associated with individuals' altruistic preferences. We contrasted the power spectrum between the more altruistic and less altruistic participants, in both the frontal region where signal correlated with other-interest and in the parietal evidence accumulation regions. No significant cluster was identified in these two regions. In addition, we focused specifically on the significant time-frequency window (~64–79 Hz and ~520–460 ms before response) in which we had found frontal-parietal synchronization (dWPLI) to be related to individual altruistic preferences. Region-of-interest analyses of oscillatory power in this time window also did not reveal any significant increases in frontal or in the parietal regions (S1 Fig). These results confirm that individuals' altruistic preferences are only related to gamma-band oscillatory synchronization between the frontal and parietal regions we studied, but not to local oscillatory power in either of these regions.

It is worth noting that the frontal and parietal areas we targeted here may also be involved in other functions that promote altruistic preferences. For instance, stronger responses in dorsal medial prefrontal cortex and TPJ have been found to be associated with representations of others' perspectives [60–62], which may in turn modulate individuals' decision weight over others' interests or altruistic preferences [8,9,12]. Nevertheless, we still believe that our results are better explained from the perspective that fronto-parietal synchronization supports the evidence integration process underlying social decisions. First, the regions we stimulate in the current study are based on our previous EEG study in which the parietal region was shown to signal evidence accumulation processes in both inequality contexts, and the frontal region was shown to synchronize with the evidence accumulation in parietal region to modulate altruistic preferences [18]. Therefore, the stimulation protocol we used was aimed to test the specific hypothesis that the gamma-band frontal-parietal synchronization, rather than local oscillation of frontal and parietal regions, is causally relevant to individuals' altruistic preferences. That is, we simultaneously modulated the oscillations of these two regions to entrain the large-scale interregional neural interactions underlying the evidence integration process [29]. Importantly, the results we found are clearly consistent with this specific a-priori hypothesis.

Second, we showed that the effects of entraining interregional phase-coupling on altruistic preferences are specific to the gamma-band frequency, rather than other frequencies such as alpha. This is clearly consistent with our previous observation of correlation between altruistic preferences and frontal-parietal phase-coupling in the gamma-band frequency [18]. In general, extracellular recording studies suggest that coordinated interaction between neural excitation and inhibition typically result in neural oscillations in the gamma-band frequency [63,64], and enhancement of phase-coupling in the gamma-band frequency has been found to facilitate efficient information transfer between distant neural populations [65,66]. These findings highlight the role of interregional gamma-band phase-coupling in carrying and transferring information signals between cortical areas, to support value integration and decision processes underlying decision-making [29].

Third, our results are also in line with previous neuroimaging studies of social decision-making, which have suggested that parietal cortex is functionally connected with other brain regions during the integration and weighting between different dimensions of information (e.g., trade-off between self-interest and other-interest) [12,24].

As a difference to previous findings, we found that tACS designed to increase frontoparietal oscillatory coherence changed the decision weight over specific choice attributes (others payoffs), rather than choice variability as observed previously for nonsocial decision-making [29]. A potential explanation for this discrepancy is that the frontal signals targeted in the two studies may have originated in different frontal regions, which may play different roles for nonsocial versus altruistic decision-making. The signals/electrode positions identified and stimulated during nonsocial decision-making [29] corresponded to the frontal pole, whereas the signals/electrode positions targeted here to increase altruism corresponded to a relatively more posterior frontal region [18]. In other brain stimulation studies of social decision-making, yet other specific prefrontal areas were targeted based on different anatomical locations identified with neuroimaging [49,67–69].

Our current field simulations indicated that the present montage produced relatively widespread current flow within the fronto-parietal network. This reduces the focal specificity of our conclusions, since the behavioral effects observed cannot be attributed to a narrowly defined subregion. Nevertheless, our use of HD-tACS represents one of the most focal

noninvasive approaches currently available for modulating long-range connectivity in specific oscillatory frequency bands, and in particular for manipulating frequency-specific phase-coupling. Moreover, although the simulated electric field may have extended partially toward the DLPFC and other frontal areas, the field strength in these areas (~0.0208–0.0330 V/m, Table 1) was markedly lower than that beneath the main electrodes (about 0.13–0.14 V/m), suggesting minimal direct modulation of DLPFC or other frontal regions activity. These findings suggest that although minor current spread may have reached adjacent frontal regions, the effective stimulation within these areas was considerably weaker than in the intended fronto-parietal target network associated with altruistic decision-making. Importantly, the stimulation parameters were identical across experimental conditions (i.e., both gamma and alpha active control), indicating that any potential off-target effects would not confound the frequency-specific nature of our results. Taken together, our results are best interpreted as evidence that HD-tACS modulated gamma-band synchronization at the network level of the fronto-parietal circuit, rather than as causal effects brought about by stimulation of a single cortical locus. Future studies building on our approach could improve spatial specificity by using individualized current modeling, neuronavigation-guided montages to target individual fMRI-defined areas, or combined TMS-EEG approaches to directly measure the neural consequences of modulating frequency-specific coherence in the fronto-parietal circuitry.

Although our results did not statistically confirm a specific role of frontoparietal coherence for altruism in different inequality contexts (i.e., a significant interaction between entrainment type and inequality context), we did observe numerically stronger effects of gamma-band enhancement on altruistic choices (model-free) and preferences (model-based) during disadvantageous inequality. Viewed formally, these findings show that information sharing between other-interest signals in frontal regions and parietal evidence accumulation is necessary for altruism in general, irrespective of inequality context. However, the specific pattern that is visually evident in our data is clearly consistent with our findings that disadvantageous inequality is accompanied by stronger other-payoff EEG signals, and that fronto-parietal coherence correlates with altruistic preference more strongly in this inequality context [18]. This conceptual similarity between the previous EEG and the present tACS results suggests that frontoparietal coherence in the gamma-band may play a particularly marked role during disadvantageous inequality.

Although previous papers have reported that people put more weight on others' payoffs in the ADV than in the DIS context [12,18,23], people still cared somewhat about others' payoffs also in the DIS context, perhaps to prevent others from gaining more profits than absolutely necessary due to "envy" emotions when observing others being better off in DIS contexts [42].

Consistent with this literature, we found in our EEG paper that people focused differentially on self- and other-payoffs in ADV and DIS, and that people's choices in the ADV context depended strongly on how much money the other person received [18]. This robust influence of other-regarding concerns onto the choice process in ADV may go hand in hand with a ceiling effect in the interregional synchronization underlying other-regarding concerns, thereby making it difficult for the stimulation to produce changes in these neural signals and the associated behavior. In contrast, people in the DIS context may focus more strongly on self-payoffs as a default, which may leave more dynamic range for the stimulation to enhance interregional coupling and lead people to incorporate other-interest signals in the decision more strongly. In turn, this may result in larger individual differences in the integration of other-regarding concern into choice process. In line with these suggestions, the current study suggests that fronto-parietal synchronization is important for people to incorporate other-interest into choice process in both ADV and DIS contexts, as the gamma entrainment effect is not completely context-specific. However, the gamma entrainment effect is more marked in the DIS context, implying that incorporation of other-interest into the choice process may vary more strongly across individuals and thus be more sensitive to the entrainment by tACS.

While a few studies have identified brain-behavior relations in the ADV [12,70], other studies concur with ours in finding specific effects in DIS [18,23,46]. Careful reading of all these papers shows that studies identifying specific effects for ADV mainly focused on structure and activity of parietal regions (i.e., TPJ) [12,70], while studies showing specific effects in DIS

mainly focused on oscillation/synchronization or functional connectivity of prefrontal regions [18,23,46], as also done here. For example, Christian and colleagues revealed that while theta oscillations in rTPJ strengthened inequality aversion generally, theta tACS over rLPFC enhanced the preference for unequal choices more strongly for disadvantageous compared to advantageous outcome distributions [46]. Together with our studies, these results thus suggest that frontal regions may play a critical role in processing of other-interest or efficiency to promote altruistic preferences in DIS.

While this proposition should be formally confirmed in future studies, it already raises interesting questions about why putting individuals in a disadvantageous position should trigger particular motives that may rely on fronto-parietal coherence. One explanation may be attentional in nature: a disadvantageous position may make people more averse to their loss relative to the outcome of the other, and may therefore motivate people to pay specific attention to the relation between their outcome and that of their advantaged partners [71,72]. This attentional sensitivity increase may correspond to a higher functional relevance of the representations of other's outcomes, which are communicated via oscillatory communication of the corresponding signals into the parietal evidence accumulation process. By contrast, participants in an advantageous position usually place a high decision weight on others' profit to start with, as demonstrated by both our modeling results and by previous studies [12]. Such a robust, constant incorporation of other-interest into choice process in the advantageous context may make it difficult to observe individual difference in the other-regarding concern, as well as in the underlying interregional synchronization. It may also reduce individuals' sensitivity to tACS entrainment of the corresponding signals into the evidence accumulation process, akin to a ceiling effect. Irrespective of such speculations, our results clearly inspire future studies to test more explicitly for differences in the role of frontoparietal coherence for different aspects of social decision-making that are thought to reflect different motives.

A limitation of the present study is that we did not directly record neural activity during the stimulation procedure. While our tACS protocol was designed to modulate fronto-parietal coherence at specific frequency bands [29], we cannot provide direct neural measures of the intended entrainment. Without concurrent EEG recordings, it remains theoretically possible that stimulation may have affected behavior also through mechanisms other than frequency-specific synchronization between the targeted regions. While technically challenging [73,74], future studies combining tACS with EEG may validate the frequency- and network-specific impact of the stimulation, to more directly link moment-to-moment changes in fronto-parietal gamma-band coherence to altruistic decision-making.

Our findings could have clinical implications, but these at present have to remain somewhat speculative: altruistic behavior is not uniformly reduced across clinical populations; rather specific components of this behavior may be atypical in different disorders. For instance, in psychopathy, atypical tendencies comprise reduced empathic concern and emotional responsiveness, which in turn can constrain prosocial or altruistic behaviors [75,76]. By contrast, in autism spectrum conditions, recent studies suggest that individuals are not inherently less altruistic; rather, difficulties often arise from challenges in mentalizing and interpreting social cues [77,78]. Similarly, in alexithymia, deficits in recognizing and labeling one's own and others' emotional states can interfere with empathic processes that support altruism [3]. In light of these distinctions, our study does not yet allow strong conclusions about specific clinical applications. While our stimulation approach may modulate frontoparietal connectivity, it remains unclear whether this would selectively influence perspective-taking, emotional resonance, or other processes underpinning altruism. We therefore propose that future work should investigate whether tACS differentially modulates distinct components of altruistic behavior across different populations. Such studies would provide evidence for more specific cognitive or affective mechanisms affected by the stimulation, which would inform clinical translation in a more targeted way.

Taken together, our results demonstrate that increasing frontoparietal coherence by means of tACS can causally enhance individuals' altruistic preferences and choices. This shows that altruistic choices have a clear neurobiological basis, which involves not just local neural computation of signals related to other's outcomes but also functional integration of these signals with other (domain-general) areas involved in response preparation. Our findings may thus also have novel implications for clinical neuroscience, as deficits in appropriate processing and integration of information

about others may impact several psychiatric and neurological disorders, including autism, psychopathy, and alexithymia [4,79,80]. Our approach may offer a promising start for developing intervention tools to improve individuals' social functions in these populations.

## Methods

### Ethics statement

All participants were informed about all aspects of the experiment and gave written informed consent before testing. The experiment conformed to the Declaration of Helsinki, and the protocol was approved by the Ethics Committee of the Canton of Zurich (KEK: 2020-01490).

### Participants

Forty-nine participants (20 females) participated in the study. All the participants were recruited from the student population of the University of Zurich and ETH Zurich, were right-handed, had normal or corrected-to-normal vision, were free of neurological or psychological disorders, and did not take medication during the time the experiment was conducted. They received between 109 and 133 CHF (depending on their choices) as compensation. Five participants were excluded from analyses because they failed to meet the requirements of the study (e.g., responding with both hands, not being able to stay awake during testing, no anatomical brain image available, and major in Psychology or Economy; remaining 44 participants: 19–32 years of age, mean 24.5 years). To determine the sample size, we followed our previous study and aimed to detect a medium effect with $d = 0.4$ [18]. Analyses with G*Power 3.1 suggested that we need 47 participants to have a power of $1 - \beta > 0.85$ to detect a medium effect with $d = 0.4$ at the level of $\alpha = 0.05$ for one-tailed $t$-test. The behavioral results in the current study (i.e., $t$-tests of the gamma (vs sham or versus alpha) entrainment effect on the altruistic choice in the disadvantageous inequality context) showed a significant gamma entrainment effect with effect size $d \approx 0.38$. Post-hoc power analyses suggested that we can achieve a power of $1 - \beta \approx 0.8$ to detect this effect ($d \approx 0.38$) with the remaining 44 participants at the level of $\alpha = 0.05$ for one-tailed $t$-test. The final power we achieved in the current study is equivalent to our previous paper (i.e., $1 - \beta > 0.80$) and many other studies in the literature [18,19,45,46,53,81,82].

### Behavioral paradigm

All testing took place in a sound proof, light-controlled room of the Laboratory for Social and Neural Systems (SNS-Lab) of the Zurich Center for Neuroeconomics (ZNE) at the University of Zurich. Each participant made 540 real decisions in a modified Dictator Game, requiring them to choose one of two allocations of monetary amounts between themselves and a receiver (another participant in the same study). We included two inequality contexts: in the DIS, the token amounts for participants were always lower than for the receiver; in the ADV, they were always higher. On each trial, one of the two options was revealed at the beginning of each trial (i.e., the reference option) and the other option was revealed at a later time (i.e., the alternative option, see below and Fig 1A for details). At the beginning of each trial, participants saw a central fixation cross together with a reference allocation option on one side of the cross. The alternative option, to be shown on the opposite side of the fixation cross, was initially hidden and replaced with "XX" symbols. After 0.5 s, the font color for all the stimuli changed from blue to green (the change direction of font color was counterbalanced across participants), indicating initiation of the trial. After a temporal jitter of 1–1.5 s (uniform distribution with a mean of 1.25 s), the alternative option was revealed (with XX symbols replaced with actual amounts). Participants had to choose the left or right option by pressing the corresponding keys on the response box with right index or middle finger within 2.5 s. The selected option was highlighted with color change. After another jitter of 0.5–2 s, the next trial started. Since all payoff stimuli were presented close to the fixation cross (i.e., visual angle smaller than 3°), participants were able to see the numbers clearly without shifting their gaze.

This sequential presentation allowed us to establish the inequality context with the presentation of the first option without having to explicitly instruct participants about the different contexts. There were 4 levels of reference options (i.e., 24/74, 24/98, 46/74, and 46/98) in each context. For the DIS context, the numbers before the slash denote the token amounts allocated to the participant and the numbers after the slash the token amount given to the partner, and vice versa for the ADV context. There were 22 or 23 alternative option levels corresponding to each reference option level; half of them were more equal than the reference option and the other half were more unequal than the reference option. The token differences for each party between the alternative and the reference options ranged from −19 to 19 (S2 Fig).

To avoid repetitions of exactly the same choices, we included three different trial sets with the same reference options and similar distributions of alternative options. The second and the third trial sets were generated by adding a random jitter (i.e., −1, +0, or +1) to self-/other-payoffs of the alternative options of the first trial set (S2 Fig). In the current study, we used the same trial sets as in our previous EEG paper [18]. This trial set comprised two types of trials: 1) unambiguous trials where choices that decrease inequality are costly (i.e., choices to maximize the other's payoff reduces one's own payoffs), and 2) ambiguous trials where choices that decrease inequality are not costly (i.e., maximizing the other's payoff does not decrease one's own payoffs). Several pilot studies showed that inclusion of both trial types is necessary to ensure people do not choose simple decision heuristics and/or only focus on one type of payoff. However, the ambiguous trials cannot be used for analyses of unambiguously altruistic behavior, since this is usually defined as actions that benefit others' interests but incur personal costs. In other words, only on unambiguous trials is there a clear conflict between self- and other-interests, so that choices can be clearly categorized as "altruistic" or "selfish" for the analyses we present in the current study. However, we still needed to keep the trial sets the same as the one in the EEG paper, to ensure that participants were in the same context and were presented with the same types of trials, to prevent heuristic or pre-determined response strategies.

For the ambiguous trials, there is no way to clearly categorize choices as altruistic or selfish. The only way to delve into these data is to use different (and conceptually weaker) definitions: for instance, choices may be defined as "partially altruistic" if the chosen options increase the other's payoff more strongly than the participant's payoff, and as "partially selfish" if the chosen option increases the participant's payoff more strongly than those of the others. Note that none of these choices involves a conflict between self- and other-interest, so that people should choose both of these option types irrespective of any altruistic concerns (since they make both themselves and the other better off). Even though people may in principle take into account this relative weighting in the increase of their own and the other's payoffs, it would be conceptually problematic (or even wrong) to pool these ambiguous trials with the unambiguous trials for analysis. Only one of the two types of trials allows clear identification of altruistic behavior as usually defined, and involves conflict between self- and other-interest.

Nevertheless, we also ran a mixed-effects analysis of the ambiguous trials (S4 Table). This confirmed that behavior is fundamentally different on these trials where altruism does not conflict with self-interest. Whereas we had observed for unambiguous trials that people (as is typical) are more altruistic in the ADV than in the DIS context, this pattern reverses for the ambiguous trials and people choose altruistically more often in the DIS than in the ADV context. These results confirm our concern that choice behavior on ambiguous and unambiguous trials is conceptually and cognitively very different and should therefore not be pooled when examining tACS entrainment effects on altruistic behavior. Thus, the ambiguous trials need to be excluded from data analyses of altruistic choices in our study.

Importantly, we note that after removing the ambiguous trials, we still have around 52 trials for each of the 6 conditions (312 trials in total). This number of trials per condition is at least similar to, and often larger than, the number of trials in other published brain stimulation studies on social decision-making [20,21,45,46,81,82]. We are therefore confident that the number of trials included in our analyses is sufficient to reliably capture the effects of tACS on (unambiguous) altruistic choice as commonly defined in the literature.

The task was divided into 6 runs, with each run lasting around 7 min. In each run, participants completed one block of the task (i.e., 30 trials, ~130 s) with gamma entrainment, one block with alpha entrainment (i.e., 30 trials, ~130 s), and two short blocks with sham stimulation (i.e., 15 trials, ~83 s for each block). The order of the blocks in each run was counterbalanced across different runs, and the two sham stimulation blocks were always interleaved with gamma and alpha entrainment blocks (to prevent possible long-lasting effects due to two consecutive entrainment blocks). After each entrainment/stimulation block, participants rated their discomfort due to the stimulation in the current block on a Likert scale ranging from 0 to 20 (0 = not uncomfortable at all, 20 = very uncomfortable). After each run, participants had an ~1-min rest. In total, the task took between 50 and 60 min, including breaks between runs.

To avoid potential attentional or visual processing bias due to the fixed position of reference/alternative payoffs, we counterbalanced stimulus positions (left versus right) trial-by-trial within participant. To lower the processing load and avoid potential response errors due to misrecognition, we fixed the position of self/other-payoffs within participants and counterbalanced it between participants.

Participation payment was determined at the end of the experiment and consisted of three parts: a base payment of 60 Swiss Francs (CHF, around $60 at the time of the experiment), one bonus (Bonus A) payment that depended on participants' own decisions, and another bonus (Bonus B) payment that was determined based on the choices of previous participants whose outcomes were allocated to the current participant. To determine these bonus payments, the participant drew two envelopes from two piles of envelopes, one pile for bonus A and one pile for bonus B. Each envelope contained five different randomly determined trial numbers.

For Bonus A, the numbers in the envelope were the trial numbers that will be selected from the full list of the participant's choices after the experiment. We calculated the mean payoff from these options and paid this as the first bonus.

For Bonus B, the numbers in the envelope were chosen from the full list of choices taken by previous participants. This list of choices was randomly drawn from the full list of choices of all previous participants, so that the participant was randomly paired with a different person on every round. The mean of the partner's payoffs across the chosen five rounds was paid out as the second bonus. The final payments derived from the two bonuses were CHF 63.9 ± 5.3 (Mean ± SD).

The partner's payoffs resulting from the participant's own choices also entered the full list of choices for future participants, meaning that they were paid out to these participants if any of the current participant's choices were selected. The exchange rate was 1 token = 0.5 CHF.

## tACS

We delivered tACS entrainment by means of a multi-channel stimulator (DC-stimulator MC, NeuroConn Ilmenau, Germany). We first applied a topical anesthetic crème (EMLA crème 5%) to the scalp location of the electrodes [83–85], which we removed after ~45 min to fix the tACS electrodes. This procedure helped improve blinding of the participants to the entrainment type (i.e., gamma/alpha tACS versus sham-tCS) and reduced the skin sensations due to tACS, thereby making the entrainment more comfortable [86].

Following our previous EEG study, we placed two arrays of small electrodes (3 × 1 electrode montage: 3 peripheral electrodes and 1 central electrode, electrode diameter: 2 cm, electrode center-to-center distance: 5 cm) to focally stimulate the frontal and parietal regions over which we had observed interregional phase coupling to correlate with altruistic behavior during disadvantageous inequality. The central electrodes were placed at the centers of the significant clusters (channel no. 37, 39, 61, 65, and 70 for the parietal region and channel no. 15, 19, 24, 28, 69, 71, 86, 87, 106, 107, 108, 112, and 113 for the frontal region on 128 channel waveguard cap, ANT Neuro, the Netherlands) identified in our previous EEG study [18]. We used a conductive paste (Ten20 EEG Conductive Paste, Weaver and Company, Colorado, USA) to fixate the electrodes on the scalp and delivered alternating currents at 72 Hz (gamma-band), modulated with a 6 Hz envelope to approximate the phase-amplitude modulation occurring endogenously in the human brain [87]. We also included a control entrainment condition in which we delivered alternating currents at 12 Hz (alpha band) modulated with a 6 Hz envelope.

For the sham stimulation condition, we delivered alternating currents as in the gamma or alpha entrainment conditions, but only for 2 s (1-s ramp-up and 1-s ramp-down phases) every 27 s during the block. The maximum peak-to-peak current intensity ranged from 2.4 mA to 4 mA, depending on each participant's tolerance for the stimulation currents (tested before the experimental runs were conducted). Participants were not aware of which type of entrainment they received in each block, or what hypotheses about the effects of entrainment on their behavior were tested.

We estimated the electric field for our tACS electrode montage by means of the SimNIBS 2.1 toolbox [88]. The estimated electric field clearly showed that the targeted parietal and frontal areas under the two main tACS electrodes are affected by the stimulation with a relatively good spatial focality (see Fig 1B).

### Electric field simulation

To clarify the specific cortical regions targeted by our tACS montage, we conducted additional analyses to map the simulated electric field distributions onto Montreal Neurological Institute (MNI) coordinates corresponding to brain regions previously implicated in other-interest processing and altruistic decision-making [9,11,12,49,50] or in "social interaction" based on Neurosynth meta-analysis. Specifically, we calculated the mean simulated electric field within 8-mm-radius spheres centered on MNI coordinates derived from (a) prior fMRI studies on altruistic and other-regarding choices, and (b) a Neurosynth meta-analysis using the term "social interaction" (Table 1) with SimNIBS [51,52].

These analyses covered key frontal regions—including the ventromedial prefrontal cortex (VMPFC: 6, 38, 9; −4, 44, −8), dorsomedial prefrontal cortex (DMPFC: 2, 56, 16), anterior cingulate cortex (ACC: 9, 36, 3), middle cingulate cortex (MCC: 1, 9, 30; 2, 26, 42), dorsolateral prefrontal cortex (DLPFC: 51, 11, 22), and inferior frontal gyrus (IFG: 52, 30, 4)— as well as parietal regions such as the temporoparietal junction (TPJ: 51, −51, 27; 63, −42, 21) and inferior parietal lobule (IPL: 57, −37, 52).

### Behavioral analysis

To examine the effects of tACS on altruistic behavior, we performed a logistic mixed-effects regression predicting trial-wise choices (altruistic = 1, selfish = 0) as a function of inequality context (DIS = 0, ADV = 1), stimulation condition, and their interaction. The stimulation condition was modeled as a categorical factor with three levels (gamma, alpha, and sham). In the lme4 package for R, a categorical predictor with $k$ levels is internally represented by $k–1$ dummy variable under treatment (reference) coding, such that each estimated coefficient reflects the contrast between one active condition (i.e., gamma entrainment) and the reference condition (i.e., alpha or sham stimulation). This approach is mathematically equivalent to the conventional dummy coding representation of a three-level categorical factor in mixed-effects models (e.g., dummy variables: gamma entrainment (gamma = 1, sham or alpha = 0); sham stimulation (sham = 1, gamma or alpha = 0); and alpha entrainment (alpha = 1, sham or gamma = 0)). The difference lies only in parameter labeling, not in the estimated model structure or statistical tests. This approach has several statistical advantages. First, logistic mixed-effects regression accounts for random subject- and item-level variability, thereby improving statistical power and generalizability compared to subject-wise estimates followed by t-tests. Second, testing contrasts within the model avoids the problem of "double dipping" that would arise if one first extracted estimates and then performed secondary tests. Third, corrections for multiple comparisons are inherently controlled when specifying orthogonal contrasts in the mixed model framework.

To test whether or not physical discomfort related to different tACS entrainment types and current intensity affected participants' choices, we also conducted regression models with each participant's discomfort rating for different types of entrainment and the individual current intensity as regressors of no interest. All mixed-effects regression models included random effects for the participant-specific intercept term. The results of these regressions are reported in Tables 2 and 3. We performed these mixed-effects regression analyses using the lme4 package in R.

Since our previous study had shown that the advantageous context enhances altruistic behavior relative to the disadvantageous context, and that greater gamma-band phase coupling is associated with stronger altruistic preferences, we

tested the directional hypotheses corresponding to these effects in mixed-effects regression models with one-tailed statistical tests. Specifically, in the regression models reported in Tables 2, 3, 4, S1, S2, and S4, we applied one-tailed statistical tests for the effects of "Inequality context" and "Gamma" entrainment, but two-tailed statistics for the other effects with undirected hypotheses. Likewise, in the post-hoc analyses contrasting altruistic choices and parameters between different conditions, we applied one-tailed t-test statistics with Bonferroni correction for multiple comparisons.

## Computational model

In the first model (M1), the subjective value difference between the more generous option and the more selfish option (VD) is constructed based on the Charness-Rabin utility model [41] in which people assign complementary weights to outcomes for themselves or others:

$$U(Generous)_{(c,s,i)} = \left(1 - \omega_{(c,s)}\right) \times G^S_{(c,s,i)} + \omega_{(c,s)} \times G^O_{(c,s,i)} \tag{1}$$

$$U(Selfish)_{(c,s,i)} = \left(1 - \omega_{(c,s)}\right) \times S^S_{(c,s,i)} + \omega_{(c,s)} \times S^O_{(c,s,i)} \tag{2}$$

$$VD_{(c,s,i)} = U(Generous)_{(c,s,i)} - U(Selfish)_{(c,s,i)} \tag{3}$$

with indices c for conditions (c = DIS and Gamma, DIS and Sham, DIS and Alpha for different entrainment conditions in disadvantageous inequality context, and c = DIS and Gamma, DIS and Sham, DIS and Alpha for different entrainment conditions in advantageous inequality context), s for participants (s = 1,..., $N_{participants}$), and i for trials (i = 1,..., $N_{trials}$). $G^S_{(c,s,i)}$ ($S^S_{(c,s,i)}$) indicates participants' payoff of the generous (selfish) option in condition c, for participant s and trial i; $G^O_{(c,s,i)}$ ($S^O_{(c,s,i)}$) indicates the partners' payoff in the generous (selfish) option in condition c, for participant s and trial i.

We calculated trial-by-trial value difference ($VD_{(c,s,i)}$) between the two offers ($VD_{(c,s,i)} = U(Generous)_{(c,s,i)} - U(Selfish)_{(c,s,i)}$) and employed a softmax function to transform these value differences into probabilities of choosing the more generous option:

$$P(Generous)_{(c,s,i)} = \frac{1}{1 + e^{-\lambda_{(s)} \times VD_{(c,s,i)}}} \tag{M1}$$

where $\lambda_{(s)}$ is participant-specific inverse temperature parameters reflecting to what extent individuals' decision depend on VD.

In the second model (M2), we included condition-specific inverse temperature parameters to examine the extent to which individuals' decision weigh on value difference across different conditions:

$$P(Generous)_{(c,s,i)} = \frac{1}{1 + e^{-\lambda_{(c, s)} \times VD_{(c,s,i)}}} \tag{M2}$$

where $\lambda_{(c,s)}$ is condition-specific free temperature parameters reflecting individuals' weight on value difference. The calculation of value difference ($VD_{(c,s,i)}$) in M2 was the same as M1.

In the third model (M3), we included a third set of parameters ($\mu_{(c,s)}$) to measure to what extent individuals' decisions weigh on the overall benefit for all parties (i.e., efficiency):

$$U(Generous)_{(c,s,i)} = \left(1 - \omega_{(c,s)}\right) \times G^S_{(c,s,i)} + \omega_{(c,s)} \times ((1 - \mu_{(c,s)}) \times G^O_{(c,s,i)} + \mu_{(c,s)} \times (G^S_{(c,s,i)} + G^O_{(c,s,i)})) \tag{4}$$

$$U(Selfish)_{(c,s,i)} = \left(1 - \omega_{(c,s)}\right) \times S^S_{(c,s,i)} + \omega_{(c,s)} \times ((1 - \mu_{(c,s)}) \times S^O_{(c,s,i)} + \mu_{(c,s)} \times (S^S_{(c,s,i)} + S^O_{(c,s,i)})) \tag{5}$$

$$VD_{(c,s,i)} = U(Generous)_{(c,s,i)} - U(Selfish)_{(c,s,i)} \tag{6}$$

$$P(Generous)_{(c,s,i)} = \frac{1}{1 + e^{-\lambda_{(s)} \times VD_{(c,s,i)}}} \tag{M3}$$

where $\mu_{(c,s)}$ is the condition-specific parameters measuring individuals' weight on efficiency and $\lambda_{(s)}$ is the participant-specific inverse temperature parameters.

We obtained best fitting parameters by maximizing the log-likelihood of the data for each model with the MATLAB function fmincon. To avoid the optimization getting stuck in local minima, we used 50 multiple starting points. To evaluate model fits, we calculated the Bayesian Information Criterion (BIC) [89], which rewards model parsimony to avoid overfitting:

$$BIC = -2\ln L + k \ln(n)$$

where $L$ is the maximized likelihood for the model, $k$ is the number of free parameters in the model, and $n$ is the number of observations.

In contrast to our original comprehensive EEG study exploring many aspects of the choice process, in the current study, we were not interested in how the gamma entrainment is causally relevant to other aspects of decision processes measured in sequential sampling models (SSM). Nevertheless, we still ran an SSM (M4) as we did in the EEG paper. However, formal model comparison suggested that the SSM model has much higher BIC than the softmax function model (i.e., M1, S3 Table).

We also followed established procedures [90] to calculate Bayes factor as $BF = \exp(-\frac{1}{2}\Delta BIC)$, where $\Delta BIC$ is the difference in the average BIC between the winning model (M1) and each alternative model. BF between 3 and 10 indicates moderate evidence, $BF > 10$ indicates strong evidence, and $BF > 100$ indicates very strong evidence that the winning model is superior to the alternative model [90].

### Parameter recovery

We used the fitted parameters from each individual participant to simulate choices with the winning model (M1) 100 times, fitted the model to these simulated choices by maximizing the log-likelihood of the data with the MATLAB function fmincon, and correlated the averaged recovered parameters (of the 100 simulations) with the ones used to simulate choices. All parameters recovered well (S5 Fig, correlations between simulated and recovered parameters $p < 0.001$ for all parameters).

### Model simulation

To clarify the relationship between the parameter of weight on other-interest and the parameter of inverse temperature in our winning model (M1), and to demonstrate that we can separate stimulation effects on these two parameters, we performed model simulations to examine and separate the effects of these parameters on choice behaviors. Systematically varying both parameters (i.e., weight on other-interest and inverse temperature) revealed that the overall probability of altruistic choice mainly increases with the weight on other-interest (left panel in S6 Fig), whereas the variance of altruistic choice mainly decreases with the inverse temperature (right panel in S6 Fig). Therefore, the softmax function model we

used can capture and separate specific effects on the weight on other-interest versus the decision precision, as expressed in qualitatively different effects in the data.

We also performed some corresponding model simulation analyses for the SSM (M4) to examine whether this more complex model would allow us to examine changes in within-trial noise due to the tACS. In these, we systematically varied the weight on other-interest and decision threshold (which corresponds directly to within-trial noise at a constant drift rate), or the weight on other-interest and drift rate. The simulation results showed a different pattern to that present for the softmax function model: the probability and variance of altruistic choice increased with weight on other-interest (S7A and S7B Fig), whereas the variance of altruistic choice decreased with both increasing decision threshold (S7A Fig right panel, upper panel for DIS and lower panel for ADV) and increasing drift rate (S7B Fig right panel, upper panel for DIS and lower panel for ADV). As the decision threshold and drift rate in the SSM have similar effects on the average altruistic choice probability and the variance of altruistic choices, fitting the SSM cannot differentiate the effects of within-trial noise (as captured by the decision threshold) and the drift rate. Thus, these simulations are consistent with the formal model comparisons, in showing that the SSM is not particularly informative about tACS effects in our specific dataset, and that our simpler model (the softmax function model) is more appropriate for capturing the potential tACS entrainment effects on decision precision versus the weight placed on other's welfare.

## Supporting information

**S1 Table. Logistic mixed-effects model results of choice data for DIS context.**
(DOCX)

**S2 Table. Logistic mixed-effects model results of choice data for ADV context.**
(DOCX)

**S3 Table. Model comparison results.**
(DOCX)

**S4 Table. Logistic mixed-effects model results of choice data for ambiguous trials.**
(DOCX)

**S1 Fig. Individual differences in altruism not relate to local oscillation of frontal or parietal region.** (A and B top panel) Heatmap showing T-statistics for the differences between the more altruistic (Gr. MA) minus less altruistic (Gr. LA) group in local power of frontal cluster (A) shown in Fig 5A of our original EEG paper and parietal cluster (B) shown in Fig 2C of our original EEG paper (https://elifesciences.org/articles/80667#fig5). No significant cluster survived cluster correction for multiple comparisons at $p < 0.05$. Moreover, no significant effect was identified in the gamma-band frequency range (~64–79 Hz) at the time window of ~520–460 ms before response (highlighted in white box) either in the frontal or in the parietal region. In contrast, the averaged dWPLI between frontal and parietal regions in this time-frequency window (~64–79 Hz and ~520–460 ms before response) correlated with individuals' altruistic preferences in our EEG paper. (A and B bottom panel) Temporal dynamics of the average power change relative to the baseline in the ~64–79 Hz frequency range. Gray shaded area indicates the duration of ~520–460 ms before response. Colored shaded areas indicate ±1 SEMs. Gr., group. Significant clusters surviving the cluster correction for multiple comparisons at $p < 0.05$ are reported, N(LA) = 19, N(MA) = 19. Gr. MA, more altruistic group; Gr. LA, less altruistic group.
(TIF)

**S2 Fig. Payoff schedule.** We included two inequality contexts: disadvantageous inequality (DIS) and advantageous inequality (ADV). In the left panel, each dot represents one allocation option and each gray line represents one pair of options that was presented to participants. Blue dots are options in DIS and pink dots are options in ADV. Dots in the

center of the circle are the 1st options and diamond dots are the 2nd options. Middle and right panels show the distributions of the self-/other-payoff changes between the 2nd and the 1st option ($\Delta S$ and $\Delta O$) in DIS and ADV, respectively. These three sets of payoff matrices (top, middle, and bottom panel) have the same reference options and similar distributions of alternative options. By having such payoff matrices of all trials, we matched self-/other-payoff differences and the resulting absolute levels of inequality across both contexts and also across the 2nd and the 1st options. This allowed us to compare choices as well as neural processing of different choice features (self- and other-payoff, inequality), between the two contexts.
(TIF)

**S3 Fig. tACS electric field density simulation for asynchronous entrainment.** Two sets of small electrodes (3 × 1 HD electrode montage) were mounted over frontal and parietal regions which were located based on our previous EEG study [18]. The normalized simulated electric field distribution showed that the targeted parietal and frontal areas under the two sets of tACS electrodes are affected by the stimulation with a poor spatial focality for asynchronous entrainment.
(TIF)

**S4 Fig. Model simulation results.** Correlations of the probability of altruistic choice between participants' true responses (*y* axis) and model simulation responses (*x* axis) based on the winning model, for DIS context (top panel) and for ADV context (bottom panel). Model simulation data are highly correlated with observed true data across all inequality contexts and stimulation types.
(TIF)

**S5 Fig. Parameter recovery.** Observed parameters (values fitted to participants' choice data in the winning model M1) are used to simulate choices, and these choices are used to recover the observed parameters. We repeated this procedure 100 times to obtain the average value of each recovered parameter and correlated the averaged recovered parameters with the observed parameters (which were used to generate the choices). Each dot plots the averaged recovered parameter from the simulated choices against the observed parameter (generating parameter), and the colored lines represent the regression fits of the observed parameters on the recovered parameters.
(TIF)

**S6 Fig. Model simulation results of the softmax function model (M1) for DIS (top panel) and ADV contexts (bottom panel).** The overall probability of altruistic choice mainly increases with the weight on other-interest for both contexts (left panel), while the variance of altruistic choice will mainly decrease with the inverse temperature for both contexts (right panel), suggesting that the two parameters in the softmax function model used here is good to capture the effects of weight on other-interest and the decision precision on altruistic choice.
(TIF)

**S7 Fig. Model simulation results of the SSM model (M4).** The probability of altruistic choice increasing with weight on other-interest (A and B left panel) while variance of altruistic choice increasing with weight on other-interest and decreasing with both decision threshold (A right panel, upper panel for DIS and lower panel for ADV) and drift rate (B right panel, upper panel for DIS and lower panel for ADV).
(TIF)

## Author contributions

**Conceptualization:** Jie Hu, Marius Moisa, Christian C. Ruff.

**Data curation:** Jie Hu, Marius Moisa.

**Formal analysis:** Jie Hu.

**Funding acquisition:** Jie Hu, Christian C. Ruff.

**Investigation:** Jie Hu, Marius Moisa, Christian C. Ruff.

**Methodology:** Jie Hu.

**Project administration:** Christian C. Ruff.

**Resources:** Christian C. Ruff.

**Supervision:** Christian C. Ruff.

**Visualization:** Jie Hu, Marius Moisa.

**Writing – original draft:** Jie Hu, Marius Moisa, Christian C. Ruff.

**Writing – review & editing:** Jie Hu, Marius Moisa, Christian C. Ruff.

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
