## [Editor Report · Decision Letter 0]

13 Jun 2025

Dear Dr Hu,

Thank you for submitting your manuscript entitled "Enhancing altruism by electrically augmenting frontoparietal gamma-band phase coupling" for consideration as a Research Article by PLOS Biology.

Your manuscript has now been evaluated by the PLOS Biology editorial staff, as well as by an academic editor with relevant expertise, and I am writing to let you know that we would like to send your submission out for external peer review.

Once your full submission is complete, your paper will undergo a series of checks in preparation for peer review. After your manuscript has passed the checks it will be sent out for review. To provide the metadata for your submission, please Login to Editorial Manager (https://www.editorialmanager.com/pbiology) within two working days, i.e. by Jun 15 2025 11:59PM.

Kind regards,

Taylor

Taylor Hart, PhD,

Associate Editor

PLOS Biology

thart@plos.org

---

## [Decision Letter · Decision Letter 1]

4 Aug 2025

Dear Dr Hu,

Thank you for your patience while your manuscript "Enhancing altruism by electrically augmenting frontoparietal gamma-band phase coupling" was peer-reviewed at PLOS Biology. It has now been evaluated by the PLOS Biology editors, an Academic Editor with relevant expertise, and by several independent reviewers who were directed to also consider the existing reviewer reports.

In light of the reviews, which you will find at the end of this email, we would like to invite you to revise the work to thoroughly address the reviewers' reports.

As you will see, the reviewers said that the results are of interest, but they noted some shortcomings in the analyses and definition of the study's limitations. Based on our discussion with the academic editor, we think that you should implement the requested analyses and textual changes. However, collecting additional data is not required.

Given the extent of revision needed, we cannot make a decision about publication until we have seen the revised manuscript and your response to the reviewers' comments. Your revised manuscript is likely to be sent for further evaluation by all or a subset of the reviewers.

**IMPORTANT - SUBMITTING YOUR REVISION**

*Re-submission Checklist*

*Published Peer Review*

*PLOS Data Policy*

*Blot and Gel Data Policy*

Sincerely,

Taylor

Taylor Hart, PhD,

Associate Editor

PLOS Biology

thart@plos.org

REVIEWS:

Reviewer's Responses to Questions

Reviewer #1: Manuscript - PBIOLOGY-D-25-01797R1

This study investigates the possibility of increasing altruistic behaviour by delivering transcranial alternating current stimulation (tACS) over frontal and parietal regions, which putatively increases fronto-parietal coherence. The authors found that applying tACS over these regions increased altruism, particularly during disadvantageous inequality. The manuscript is clear and well-written; the study is well-thought and methodologically sound. The computational models are well formulated, stable and robust across the parameter ranges. I do, however, have a few comments and suggestions that I detail below.

Major points:

- Introduction: the authors have written a clear and eloquent introduction, including the relevant literature that nicely builds up towards the experimental questions. I do, however, miss the hypothesis / predictions of the results at the end for the introduction, e.g. after the research questions.

- Increasing interregional synchronisation with tACS: the authors describe a clear tACS procedure in which they putatively entrain interregional coherence between the frontal and parietal areas at specific frequency bands - i.e. gamma and alpha. Although, it is likely that a tACS protocol tuned at these specific frequency bands lead to an entrainment of such frequency bands during the task, a different (alternative) impact of the tACS on EEG oscillatory synchrony cannot be ruled out. This is, it is possible that tACS' effect on interregional communication between the areas targeted might not have had the expected effect. It would have been beneficial to directly measure the effects of tACS on interregional coherence with EEG recordings, at least for a subset of the participants. The examination of the EEG recordings before vs during (or after) the tACS intervention would have revealed the direct (causal) impact of the tACS in fronto-parietal coherence, significantly strengthening the interpretation of the results. It would be advisable that the authors are able to show in a subset of participants the direct effect of frequency-specific tACS on EEG coherence between the frontal and parietal regions. At the very least, the authors should acknowledge this limitation in the discussion and interpretation of the results.

Minor points:

- Abstract: the abstract mention something about EGG findings. I wonder if the authors mean EEG findings.

- Introduction: the authors nicely explain the concept of inter-regional coherence and how this can be modulated with different NIBS protocols. I think the introduction would benefit from including other NIBS protocols that have been used to selectively manipulate intra-areal coherence testing the Communication Through Coherence Hypothesis - for example: Trajkovic, J., Romei, V., Rushworth, M. F., & Sel, A. (2023). Changing connectivity between premotor and motor cortex changes inter-areal communication in the human brain. Progress in Neurobiology, 228, 102487.

Reviewer #2: This paper investigates the causal role of gamma rhythms within the frontoparietal network in encoding other-related prosocial acts - altruistic behaviors. The study uses an elegant experimental design that effectively measures altruistic behavior, building upon prior EEG findings that linked frontoparietal gamma frequencies with altruistic tendencies. By using tDCS stimulation to modulate gamma rhythms in the frontoparietal network, the authors demonstrate a causal involvement of these frequencies in altruistic acts in the context of disadvantageous inequity. The precise mechanisms underlying altruistic strategies were detected by applying computational modeling, demonstrating the causal role of gamma rhythms in assigning weight to other-regarding concerns during the decision-making process. This work is of significant interest for scientific research in cognitive neuroscience and neuroeconomics due to its novel findings on the causal manipulation of altruistic behavior.

However, the current work is not yet ready for publication and requires substantial revisions to strengthen its findings and interpretations.

Major Revisions

The current behavioural modeling approach lacks sufficient clarity and must be simplified:

(1) Table 1 and the description suggest that tDCS conditions (gamma, alpha, sham) were entered as separate binary predictors. It would be more appropriate and interpretable to model the tDCS condition as a single, three-level categorical factor, given that both sham and alpha serve as control conditions. A model structure including Inequality_Type x tDCS_conditions as regressors would allow for a more direct test of whether gamma stimulation differs from both sham and alpha stimulation, thereby strengthening the specificity and interpretability of the findings. It is essential that the effect of gamma frequencies is not only significant in contrast to sham but also significantly different from alpha, which needs to be reflected in the LMER model results by a significant interaction effect of Inequality_Type x tDCS_conditions.

(2) For a further understanding of how these three conditions (gamma, alpha, sham) are related to each other, a post-hoc pairwise comparison between the three tDCS levels (e.g., gamma vs. sham, gamma vs. alpha, alpha vs. sham) should be reported, applying appropriate corrections for multiple comparisons. Based on these regression models, as previously described by the authors, one-tailed/ two-tailed statistical tests can be calculated (e.g., by extracting individual estimates from these models).

(3) The results of the LMER models and post-hoc analyses should be presented in corresponding tables. It is not necessary to include model outputs with control variables such as discomfort in the tables; these can be briefly summarized in the Results section to demonstrate the robustness of the main findings.

Limitations of tDCS Montage and Source Analysis:

(1) Based on the SimNIBS simulations, the frontal-parietal network that was targeted looks quite widespread. This should be described as a limitation in the discussion, as it reduces the focal specificity of the findings.

(2) The authors describe the tDCS montage based on the area where gamma rhythms during the altruistic choice process were recorded, building on their previous EEG results. Nonetheless, as EEG has low spatial resolution and signals can be recorded in one region while originating from another, it still remains essential to specify which exact areas within the frontal and parietal cortices were targeted by tDCS. It would be particularly informative to assess whether the stimulated regions align with areas previously implicated in prosocial and altruistic decision-making (i.e., those involved in weighing other-related concerns), using MNI coordinates reported in prior studies. Linking the present stimulation sites to such findings would help clarify the specific type of altruistic behavior influenced by brain stimulation. For example, it would be relevant to examine whether stimulation targets part of the ventromedial prefrontal cortex, which has been previously associated with promoting other-regarding preferences (Hutcherson et al., 2015). Such clarification would also strengthen the novelty of the findings by highlighting how tDCS, in contrast to previous EEG results, contributes to understanding the focal specificity of frontal brain regions involved in promoting altruistic choices.

(3) The SimNIBS simulation suggests that tDCS stimulation might have targeted adjacent brain regions, such as the Dorsolateral Prefrontal Cortex (DLPFC), in addition to the fronto-parietal network linked to altruistic behavior. Given the ongoing debate about the DLPFC's role in social decision-making, particularly whether it promotes prosocial or selfish tendencies in social decision-making (e.g., presence of punishment; Knoch et al., 2006; Buckholtz et al., 2013) with current evidence suggesting that rDLPFC does not promote prosocial giving (Christian et al., 2022), I would recommend calculating an additional source localization analysis to assess whether tDCS stimulation targeted adjacent areas such as the DLPFC, and if necessary, discuss this shortly as a further limitation.

Minor Revisions

(1) The manuscript repeatedly refers to potential clinical implications of modulating frontoparietal connectivity for promoting altruistic behavior, particularly for autism or alexithymia. For example, the statement, "Experimentally clarifying the causal role of frontoparietal connectivity for altruism may be a necessary step towards developing and/or improving intervention approaches to facilitate altruism, e.g., in people affected by social apathy and other related psychiatric and neurological disorders," is overly general and somewhat outdated, especially regarding individuals with autism. The term "social apathy" lacks clinical specificity and requires a more precise description based on current clinical literature. Moreover, recent research (e.g., Okuzumi, 2024) challenges the assumption that individuals with autism are inherently less altruistic than neurotypical individuals. Instead, difficulties in interpreting and responding to others' social cues—rather than a reduced capacity for altruism itself—are more accurate characterizations of social challenges in autism and alexithymia. Thus, I recommend that the authors specify which components of altruistic behavior may be atypical in these populations (e.g., perspective-taking) and clarify how tDCS might modulate these particular processes. More importantly, I suggest weakening or removing claims about clinical relevance or therapeutic applications, as enhancing altruism per se may be less meaningful than addressing the underlying mechanisms of social understanding that influence prosocial behavior.

(2) In the introduction, the authors describe the role of frontal brain areas in prosocial decision-making in broad terms, without specifying the precise role of distinct frontal brain regions in self- versus other-oriented behavior. This weakens the theoretical clarity of the manuscript. Given that the role of the prefrontal cortex for social decision-making is highly context-dependent, it is important that the authors describe this in more detail which distinct parts of the prefrontal cortex are involved or causally relevant in promoting selfish or other-related interests.

---

## [Decision Letter · Decision Letter 2]

16 Dec 2025

Dear Dr Hu,

Thank you for your patience while we considered your revised manuscript "Enhancing altruism by electrically augmenting frontoparietal gamma-band phase coupling" for publication as a Research Article at PLOS Biology. This revised version of your manuscript has been evaluated by the PLOS Biology editors, the Academic Editor, and the previous reviewers.

Based on the reviews, we are likely to accept this manuscript for publication. Please also make sure to address the following data and other policy-related requests.

IMPORTANT: Please ensure that your next revision addresses the following editorial points:

-----------------

**Title:

We would like to tweak your study's title to make it read as less 'passive', in accordance with our stylistic preferences. Is the following reformulation acceptable to you?

Augmentation of frontoparietal gamma-band phase coupling enhances human altruistic behavior

**Ethics:

-- The Ethics statement needs to be a separate, independent (and the first) subheading in the Material & Methods section.

**Data:

-- Thank you for uploading your data and code to OSF. We see that you have provided .csv and Matlab files containing the data. We would also appreciate if you could add a supplementary file containing the numerical data underlying the plots in Figure 2, so that users are not required to run the code to examine this data. You can either add this as a supplementary information file called "S1 Data" (upload as "S1_Data.xlsx") or include it in your online deposition. Either way, please cite the location of the data clearly in all relevant main and supplementary Figure legends, e.g. “The data underlying this Figure can be found in "URL”

-- We see that you have uploaded a supplementary file containing additional methods details, together with the supplementary tables and figures. As supplementary document files are rarely read and are not proofread, we prefer if you could integrate all of the methods information into the main methods files, along with the tables, and upload the supplementary figures individually as supporting information.

-----------------

We expect to receive your revised manuscript by January 5.

*Published Peer Review History*

*Press*

Sincerely,

Taylor

Taylor Hart, PhD,

Associate Editor

thart@plos.org

PLOS Biology

Reviewer remarks:

Reviewer #1: The authors have now addressed all my comments. I am happy to support the publication of the manuscript in its current format

Reviewer #2: the authors have addressed my concerns in detail and that I recommend the adapted version of the manuscript for publication. I have no further comments or requests for a second revision.

---

## [Editor Report · Decision Letter 3]

6 Jan 2026

Dear Dr Hu,

Thank you for the submission of your revised Research Article "Augmentation of frontoparietal gamma-band phase coupling enhances human altruistic behavior" for publication in PLOS Biology. On behalf of my colleagues and the Academic Editor, Raphael Kaplan, I am pleased to say that we can in principle accept your manuscript for publication, provided you address any remaining formatting and reporting issues. These will be detailed in an email you should receive within 2-3 business days from our colleagues in the journal operations team; no action is required from you until then. Please note that we will not be able to formally accept your manuscript and schedule it for publication until you have completed any requested changes.

PRESS

Sincerely,

Taylor

Taylor Hart, PhD,

Associate Editor

PLOS Biology

thart@plos.org